# The Importance of Abnormal Platelet Count in Patients with *Clostridioides difficile* Infection

**DOI:** 10.3390/jcm10132957

**Published:** 2021-06-30

**Authors:** Shira Buchrits, Anat Gafter-Gvili, Jihad Bishara, Alaa Atamna, Gida Ayada, Yair Eynath, Tomer Avni

**Affiliations:** 1Internal Medicine Department A, Rabin Medical Center, Beilinson Hospital, Petah-Tikva 49100, Israel; gaftera@gmail.com (A.G.-G.); yair.ey@gmail.com (Y.E.); boskko2001@gmail.com (T.A.); 2Sackler Faculty of Medicine, Tel Aviv University, Tel-Aviv 69978, Israel; prof.jbishara@gmail.com (J.B.); alladinat@clalit.org.il (A.A.); 3Infectious Disease Unit, Rabin Medical Center, Beilinson Hospital, Petah-Tikva 49100, Israel; 4Internal Medicine Department C, Rabin Medical Center, Beilinson Hospital, Petah-Tikva 49100, Israel; ghidaa.692@gmail.com

**Keywords:** thrombocytosis, thrombocytopenia, mortality, *Clostridioides difficile*

## Abstract

Background: *Clostridium difficile* infection (CDI) causes morbidity and mortality. Platelets have been increasingly recognized as an important component of innate and adaptive immunity. We aimed to assess the incidence of thrombocytopenia and thrombocytosis in CDI and the effect of an abnormal platelet count on clinical outcomes. Methods: This single-center, retrospective cohort study consisted of all adult patients hospitalized in Rabin Medical Center between 1 January 2013 and 31 December 2018 with laboratory confirmed CDI. The primary outcome was 30-day all-cause mortality. Risk factors for 30-day all-cause mortality were identified by univariable and multivariable analyses, using logistic regression. Results: A total of 527 patients with CDI were included. Among them 179 (34%) had an abnormal platelet count: 118 (22%) had thrombocytopenia and 61 (11.5%) had thrombocytosis. Patients with thrombocytosis were similar to control patients other than having a significantly higher white blood cell count at admission. Patients with thrombocytopenia were younger than control patients and were more likely to suffer from malignancies, immunosuppression, and hematological conditions. In a multivariable analysis, both thrombocytosis (OR 1.89, 95% CI 1.01–3.52) and thrombocytopenia (OR 1.70, 95% CI 1.01–2.89) were associated with 30-days mortality, as well as age, hypoalbuminemia, acute kidney injury, and dependency on activities of daily living. A sensitivity analysis restricted for patients without hematological malignancy or receiving chemotherapy revealed increased mortality with thrombocytosis but not with thrombocytopenia. Conclusions: In this retrospective study of hospitalized patients with CDI, we observed an association between thrombocytosis on admission and all-cause mortality, which might represent a marker for disease severity. Patients with CDI and thrombocytopenia also exhibited increased mortality, which might reflect their background conditions and not the severity of the CDI. Future studies should assess thrombocytosis as a severity marker with or without the inclusion of the WBC count.

## 1. Background

Clostridioides difficile infection (CDI) is the most common healthcare-associated infection in the United States [1,2]. Older age and increased comorbidity are risk factors for CDI. The rate of CDI among patients older than 65 years is 20 times higher than in younger individuals. Increased comorbidity is also related with CDI, partly by its association with greater contact with health-care and need for hospitalization [3]. There are variety of severity scores that asses the severity of CDI early in the course of the disease, based on expert opinion, which includes leukocytosis, low albumin levels, and creatinine levels or acute kidney injury [4,5,6]. Given that thrombocytopenia is a poor prognostic risk factor for some systemic infections, several studies hypothesized that thrombocytopenia may have prognostic significance in CDI [7]. While thrombocytopenia has been most frequently associated with increased mortality in sepsis, the impact of thrombocytosis on clinical outcomes remains underreported [8]. We aimed to assess the incidence of thrombocytopenia and thrombocytosis in CDI and the effect of an abnormal platelet count on clinical outcomes.

## 2. Materials and Methods

This was a single center, retrospective, cohort study. The study took place at Rabin Medical Center, Beilinson Hospital, a 900primary and tertiary care university hospital. Data were collected from electronics records of all follow-up adult patients, hospitalized in all medical wards between January 2010 and December 2018, with laboratory confirmed CDI. Patients hospitalized in surgical wards, surgical ICUs, obstetrics, and gynecology wards and outpatient clinics, were excluded. CDI was defined as clinical symptoms (abdominal pain, fever, 3 or more unformed stools for day), and either positive C.DIFF QUIK CHEK COMPLETE assay for Clostridioides difficile or Xpert C.difficile PCR positive assay for C.difficile toxins. We excluded patients with a documented chronic abnormal platelet count in conditions as idiopathic thrombocytopenic purpura or essential thrombocytosis and other myeloproliferative disorders. Patients with thrombocytopenia due to active malignancy (with or without chemotherapy) were not excluded.

### Data Collection and Definitions

The data were collected through our hospital’s electronic patient computer file. Data collection includes demographic data, chronic comorbidities, age-adjusted Charlson’s comorbidity index, and immunosuppression status. Additionally collected were: data about the current infection, date and type of positive test for *Clostridioides difficile*, vital signs at admission, laboratory results (albumin, creatinine, blood count, inflammatory markers), medications at baseline, antibiotics given to treat CDI, intensive care unit (ICU) admission, length of hospital stay (LOS), and requirement for surgical procedure (colectomy). Thrombocytopenia was defined as a platelet count below 150,000/microL. The degree of thrombocytopenia was be subdivided into mild (100–150 K/microL), moderate (50–99 K/microL), and severe (<50 K/microL). Thrombocytosis was defined as a platelet count above 450,000/microL. We defined severe thrombocytosis as a platelet count above 1,000,000/microL. Recurrence of CDI (rCDI) was defined as a repeat positive CDI episode occurring within 30 days after completion of treatment and discontinuation of CDI index symptoms. Hypoalbuminemia was defined as albumin levels ≤ 3.0 mg/dL. Acute kidney injury (AKI) was defined following the KDIGO guidelines as an increase in serum creatinine by ≥0.3 mg/dL within 48 h, or an increase in serum creatinine to ≥1.5 times baseline, which is known or presumed to have occurred within the prior seven days, or urine volume <0.5 mL/kg/hour for six hours [9].

## 3. Outcomes

The primary outcome of the study was 30-day all-cause mortality. Secondary outcomes included 90-day all-cause mortality, LOS, rates of rCDI within 90 days, re-hospitalization for any cause within 90 days, need for colectomy, need for ICU transfer or admission, and length of diarrheal disease.

## 4. Statistical Analyses

The analysis was done by SPSS Statistics 25. Statistical significance was established at two-tailed comparison with *p* < 0.05. We expected a higher mortality rate in patients with an abnormal platelet count compared to controls. Assuming a 30-day mortality rate of 25% in the controls, and over 35% for the exposed population, given a patient rate of 1:4 (abnormal/normal platelet count), we needed a sample of at least 352 controls and 88 patients with abnormal platelets (sample size for proportions, alpha = 0.05, power = 0.8, 2-sided). Data were compared between the control group and thrombocytopenia and thrombocytosis separately. Dichotomous outcomes were compared using the Pearson χ^2^ test. Continuous variables were compared using T-test or the Mann-Whitney U test, as appropriate. We established risk factors for the primary outcome by univariable analysis followed by multivariable analysis. Thrombocytopenia, thrombocytosis and variables significantly associated with the primary outcome on univariable analysis, or deemed clinically relevant were entered into a multivariable logistic regression. Odds ratios (OR) with 95% confidence intervals (CI) were calculated. Pre-defined subgroup analysis was performed for patients with malignancy, immunosuppression, and per the degree of the thrombocytopenia/thrombocytosis. As a sensitivity analysis, we excluded patients with hematological malignancy, stem cell transplantation and patients receiving active chemotherapy and repeated our analysis for the primary outcome.

## 5. Results

A total of 527 patients with CDI were enrolled. Among them 179 (34%) had an abnormal platelet count: 118 (22%) had thrombocytopenia and 61 (11.5%) had thrombocytosis. Baseline characteristics of patients, according to the platelet counts group are shown in Table 1. Among the thrombocytopenic patients, the mean platelet count was 90.2 ± 42 K/microL vs. 267 ± 42 K/microL, *p* = 0.001 for the control. In 48% (57/118) thrombocytopenia was mild, moderate in 30% (35/118), and severe in 22% (26/118) of patients. At the index point, patients with thrombocytopenia were more likely to be younger (mean age 65.5 ± 16.6 years vs. 71.8 ± 17 years), of male gender (68/118 (57.6%) vs. 150/348 (43.1%), *p* = 0.013), more likely to be independent in their activities of daily living (ADL), (74/118 (62.7%) vs. 15/61 (24.5%), *p* = 0.000), more likely to suffer from hematological malignancy (34/118 (28.8%) vs. 22/348 (6.32%)), to have undergone solid organ transplantation (18/118 (15.2%) vs. 23/348 (6.6%), *p* = 0.01) and stem cell transplantation (12/118 (10.1%) vs. 11/348 (3.1%), *p* = 0.000), to be receiving chemotherapy (33/118 (27.9%) vs. 31/348 (8.9%), *p* = 0.00), and immunosuppressed (68/118 (57.6%) vs. 66/348 (18.9%), *p* = 0.00).

Among patients with thrombocytosis, the mean platelet count was 569 ± 122 K/microL vs. 267 ± 42 K/microL, *p* = 0.001 for the control. No patient had severe thrombocytosis. Patients with thrombocytosis were similar to the patients with normal platelet counts in their baseline characteristics. However, they were more likely to have coronary heart disease (18/61 (29.5%) vs. 60/348 (17.2%), *p* = 0.05). Patients with thrombocytosis were less likely to be immunosuppressed (4/61 (6.5%) vs. 66/348 (18.9%), *p* = 0.018). Other chronic conditions and the age-adjusted Charlson’s comorbidity index were similar.

The diagnosis of CDI was made by PCR in 130/348 (37.3%) of control patients and in 25/61 (41%) of patients with thrombocytosis, but much more among thrombocytopenic patients 72/118 (61.1%), *p* = 0.008. Vital signs (blood pressure, temperature, room-air saturation and pulse) at the index point were similar among all patients. Laboratory tests showed that blood hemoglobin, CRP, creatinine and albumin were similar between the groups. Total white blood cell (WBC) count was significantly increased in the group of thrombocytosis compared to the control, 17.8 ± 8.6 K/microL vs. 8.6 ± 16.5 K/microL *p* = 0.016, while patients with thrombocytopenia had significantly decreased WBC compared to the control (8.6 ± 16.5 K/microL vs. 12.8 ± 7.2 K/microL, *p* = 0.042). The frequency of patients with AKI was similar between patients with thrombocytopenia, thrombocytosis, and the control (34/118 (28.8%) vs. 14/61 (22.9%) and 97/348 (27.8%)), as well as, blood albumin levels (3.26 ± 0.68 mg/dL vs. 2.9 ± 0.57 mg/dL, and 3.28 ± 0.66 mg/dL, *p* value nonsignificant for all comparisons). Overall, 47% of the patients were treated with vancomycin (252/527), without significant differences between the groups, while 50% (267/527) were treated with metronidazole, without significant differences between the groups. Only 5/527 (0.9%) of patients were treated with fidaxomicin.

## 6. Outcome and Complications

### Mortality

The clinical outcomes are summarized in Table 2. Total 30-day all-cause mortality for the entire cohort was 30.7% (162/527). Mortality rates were 30.5% (36/118, *p* = 0.6) for patients with thrombocytopenia, 47.5% (29/61, *p* = 0.02) for patients with thrombocytosis and 27.8% (97/348) for control patients. Overall mortality for 30 days was significantly higher in patients with thrombocytosis (OR 1.7, 95% CI 1.03–2.8), and similar between thrombocytopenic patients and the control (OR 0.99, 95% CI 0.64–1.51). Univariable analysis for 30-days mortality is presented in Appendix A. Risk factors for mortality included age, ADL status, dementia, congestive heart failure (CHF), chronic kidney disease (CKD), COPD, thrombocytosis, hypoalbuminemia, and AKI, without a significant association between mortality and thrombocytopenia, (OR 0.98, CI 0.63–1.53). In a multivariable analysis, the following risk factors were associated with 30-days mortality (Table 3): age, hypoalbuminemia, AKI, ADL status (dependent), thrombocytosis (OR 1.89, 95% CI 1.01–3.52). The introduction of thrombocytopenia (of any degree) into the multivariable regression model was significantly associated with mortality (OR 1.70, 95% CI 1.01–2.89). Subgroup analysis of immunosuppressed patients and patients with active malignancy showed no increased mortality with thrombocytopenia or thrombocytosis. Mortality rates were similar between all subgroups of thrombocytopenia according to the degree: (31.5% (18/57) for mild, 28.5% (10/35) for moderate, and 30.7% (8/26) for severe, *p* = 0.9). A sensitivity analysis restricted to patients without hematological malignancy, stem cell transplantation and patients receiving active chemotherapy, revealed mortality rates of 37.3% (83/222) for the control, 40%, 19/47 for thrombocytopenia group (*p* = 0.07), and 50% 29/58 for the thrombocytosis group (*p* = 0.02). The all-cause 90-day mortality rate was 44% (52/118) for patients with thrombocytopenia, 62.2% (38/61) for patients with thrombocytosis and 35.9% (125/348) for patients with a normal platelet count. Ninety-day all-cause mortality was significantly higher among patients with thrombocytosis (OR1.73, 95% CI, 1.1–2.73) in a univariable analysis, but not for patients with thrombocytopenia (OR 1.1, 95% CI 0.76–1.6).

## 7. Other Outcomes

The rate of admission or transfer to ICU was higher in those with thrombocytopenia 5/118 (4.2%) compared with 1/61 (1.61%) for patients with thrombocytosis and 6/348 (1.7%) for the control, but the difference was not statistically significant. The diarrhea duration was similar between the groups. The LOS for patients discharged alive, was similar between the groups: 20.5 ± 20.3 days for patients with thrombocytosis, 19.1 ± 31 days for patients with thrombocytopenia, and 20.1 ± 20.5 days for the control group. There was no difference in the rates of rehospitalization, for patients with thrombocytopenia 26.2% (31/118) and for patients with thrombocytosis 24.5% (15/61) and the control 24.4% (85/348). The rate of rCDI, was 21/348 (6.0%) for the control, highest among patients with thrombocytopenia 12/118 (10.1%), and lowest among patients with thrombocytosis 1/61 (1.6%), however, the difference did not reach statistical significance due to the low number of rCDI events (*p* = 0.08).

## 8. Discussion

In this retrospective cohort study, we demonstrated that abnormal platelet counts on admission are associated with adverse clinical outcomes for hospitalized patients with CDI. Thirty-day all-cause mortality was significantly higher among patients with thrombocytosis (OR 1.89, 95% CI 1.01–3.52) and patients with thrombocytopenia (OR 1.70, 95% CI 1.01–2.89). The negative impact of thrombocytosis on mortality was also significant 90 days from the index test (OR 1.73, 95% CI, 1.1–2.73). Other risk factors for mortality included previously recognized risk factors such as age, dementia, hypoalbuminemia, AKI, and CHF. We did not demonstrate statistically significant association between abnormal platelet count on other outcomes as ICU admission, increased LOS or rCDI.

Based on our results, it could be assumed that patients with abnormal platelet counts are different from those with normal counts in several important features. Patients with thrombocytosis were similar to the control patients in their baseline characteristics and comorbidities, but presented with significantly more severe disease, as demonstrated by increased WBC counts, a well-known marker of severity [10]. We assume the mechanisms affecting the outcomes of CDI patients with thrombocytosis may probably be attributed to the severity of the disease and the inflammation (since the platelet is regarded as an acute phase reactant). Thus, the increased 30-day and 90-day mortality may probably be attributed to the severity of the disease. While on the other hand, patients with thrombocytopenia represented a different group, composed of younger patients with high rates of immunosuppression, solid and hematological malignancies, organ transplantations, and hepatic diseases. Notably, patients with thrombocytopenia were more likely than the other groups to be diagnosed by stool PCR and not by C.DIFF QUIK CHEK COMPLETE assay, thus possibly representing milder CDI disease based on several cohort studies [11,12]. Our patients with thrombocytopenia had a different clinical course with CDI and a higher mortality rate on multivariable analysis. However, this might be attributed to the severity of their comorbidities rather than that of the CDI, as suggested by our sensitivity analysis (excluding patients with disease or chemotherapy related platelet count abnormalities), which showed a non-significantly increased risk of mortality with thrombocytopenia (37% vs. 40%, *p* = 0.07).

Platelets have been increasingly recognized as an important component of innate and adaptive immunity. Thus, the contribution of blood platelets to sepsis pathophysiology has been the subject of renewed attention [13,14]. The relationship between an abnormal platelet count and CDI has been incompletely described in the literature. In a single center retrospective study of 533 patients diagnosed with CDI, 30.2% of patients were thrombocytopenic at presentation (platelet count <1,500,000). In a multivariable analysis, moderate thrombocytopenia was associated with severe disease but not with mortality [7]. In another single-center retrospective study, thrombocytopenia (<150,000 K/mL) was associated with mortality in concert with our results. However, this was only in a univariable analysis [15]. Data regarding the importance of thrombocytosis on sepsis in CDI are sparse. In a study of 162 patients with CDI, 22% had thrombocytosis on admission, but the clinical significance was not described [10]. In another retrospective analysis of six US hospitals, both thrombocytopenia and thrombocytosis on admission were associated with hospital-acquired CDI, but there were no data regarding the CDI course itself [16].

Patients suffering from malignancy have an increased incidence of CDI, with increased rates of mortality, clinical failure, rCDI, and length of hospitalization [17]. This may be attributed to several mechanisms that include: increased exposure to broad-spectrum antibiotics; chemotherapy regimens may act as antibacterial [18]; 5-fluorouracil inhibits proliferation as aerobic intestinal bacteria [19]; direct mucosal toxicity and impaired repair ability with topoisomerase inhibitors [18], and platinum-based agents [19,20]; older age [21]; and usage of proton pump inhibitors [22].

This is the first study that assessed the association between thrombocytosis on admission and the clinical course of CDI. We used multiple regression models to assess the effect of the platelet counts on mortality, which were consistently positive for the association between thrombocytopenia and thrombocytosis and 30-day mortality. Our results also showed a significant association between previously well-known risk factors for mortality in sepsis and especially risk factors for mortality in CDI, such as age, albumin levels, and AKI, thus confirming the validity of our results.

Our study has several limitations that merit consideration, these include: this was a single center cohort study, and the results might not be applicable to other centers. Second, this was an observational retrospective cohort, thus the decision to test or treat patients based on their test for CDI was under the sole discretion of their treating physician. Third, we might not have accounted for other factors associated with mortality in CDI that were not apparent by the covariate analysis described previously.

Our study sheds new light regarding the prognostic significance of thrombocytosis in CDI. We demonstrated that patients with thrombocytosis had severe disease and increased mortality. Future studies that will incorporate thrombocytosis or any abnormal platelet counts onto the CDI severity score, might better reflect short-term clinical outcomes and need for treatment for severe disease than currently existing severity markers.

In conclusion, in this study of hospitalized patients with CDI, we observed an association between thrombocytosis and all-cause mortality. Patients with CDI and thrombocytopenia also exhibited increased mortality, which might reflect their background conditions and not the severity of the CDI itself.

## Figures and Tables

**Table 1 jcm-10-02957-t001:** Characteristics of patients.

		Thrombocytopenia *N* = 118	Thrombocytosis *N* = 61	Control *N* = 348	*p* Value Thrombocytopenia/Thrombocytosis
**Demographics and Underlying Conditions**					
Age		65.5 ± 16.6	73.72 ± 19.1	71.8 ± 17	0.000/0.76
Female %		50/118 42.3%	37/61 60.1%	198/348 56.8%	0.006/0.58
ADL status–Dependent		74/118 62.7%	15/61 24.5%	144/348 41.3%	0.000/0.013
Housing–Home		109/118 92.3%	41/61 67%	269/348 77.1%	0.000/0.9
Dementia		3/118 2.5%	8/61 13.1%	33/348 9.4%	0.015/0.38
Charlson’s Score		5.78 ± 2.9	6.00 ± 24	6.08 ± 2.6	0.07/0.65
DM		28/118 23.7%	19/61 31.4%	103/348 29.5%	0.22/0.80
IHD		12/118 10.1%	18/61 29.5%	60/348 17.2%	0.065/0.04
CHF		17/118 14.4%	13/61 21.3%	60/348 17.2%	0.47/0.44
CVA		9/118 7.6%	11/61 18.1%	43/348 12.3%	0.159/0.22
COPD		7/118 5.9%	3/61 4.9%	26/348 7.47%	0.57/0.47
CKD		13/118 11%	5/61 8.1%	39/348 11.2%	0.95/0.48
Immune Suppression-any		68/118 57.6%	4/61 6.5%	66/348 18.9%	0.000/0.018
Malignancy-any		64/118 54.2%	12/61 19.6%	68/348 19.5%	0.000/0.98
Malignancy solid		12/118 10.1%	10/61 16.3%	19/348 5.4%	0.07/0.06
Solid organ transplant		18/118 15.2%	0/61 0%	23/348 6.6%	0.01/0.017
Haemato-oncology malignancy		34/118 28.8%	2/61 3.27%	22/348 6.32%	0.000/0.35
SCT		12/118 10.1%	0/61 0%	11/348 3.1%	0.000/0.15
Receiving chemotherapy		33/118 27.9%	2/61 3.2%	31/348 8.9%	0.000/0.13
Liver Disease		6/118 5%	0/61 0%	2/348 0.57%	0.01/0.55
**Vital Signs and Laboratory at Presentation**					
CDI Diagnosis method-PCR		72/118 61.1%	25/61 41%	130/348 37.3%	0.006/0.12
SBP (mmHG)		117.8 ± 23	119.2 ± 25	122.8 ± 25	0.53/0.82
Pulse		94.1 ± 20	95.0 ± 18	88.5 ± 18.5	0.21/0.69
RA saturation		96.2 ± 3.4	96.1 ± 4.2	95.4 ± 4.9	0.056/0.22
Temperature		37.1± 0.78	37.2± 0.87	37.0± 0.75	0.91/0.45
WBC (10^9^/L)		8.6 ± 16.5	17.8 ± 8.6	12.8 ± 7.2	0.016/0.042
HGB (g/dL)		10.1 ± 2.2	10.0 ± 2.1	11.1 ± 2.0	0.54/0.59
PLT		90.2 ± 42	569 ± 122	267 ± 78	0.001/0.001
Creatinine		1.28 ± 0.79	1.12 ± 0.84	1.29 ± 1.07	0.39/0.34
CRP (mg/L)		10.12 ± 10.9	10.7 ± 9.2	9.79 ± 9.21	0.07/0.95
Albumin (g/dL)		3.26 + 0.68	2.9 ± 0.57	3.28 ± 0.66	0.32/0.188
PLT count abnormality Degree	Severe Moderate Mild	26/118 22% 35/118 30% 57/118 48%	0/61 0% 61/61 100%	NA	NA
AKI		34/118 28.8%	14/61 22.9%	97/348 27.8%	0.68/0.75
Hypoalbuminemia		46/118 38.9%	36/61 59%	136/348 39.0%	0.98/0.004

Abbreviations: AKI acute kidney injury; CHF Congestive heart failure; COPD Chronic obstructive pulmonary disease; Ckd Chronic kidney disease; CVA cerebrovascular events; DM diabetes mellitus; HGB Hemoglobin; IHD Ischemic heart disease; PLT Platelet Count; SCT Stem cell transplantation; SBP Systolic blood pressure; WBC White blood count.

**Table 2 jcm-10-02957-t002:** Clinical Outcomes.

	Thrombocytopenia *N* = 118	Thrombocytosis *N* = 61	Control *N* = 348	*p* Value Thrombocytopenia/Thrombocytosis
Mortality 30 days	36/118 30.5%	29/61 47.5%	97/348 27.8%	*p* = 0.87/ *p* = 0.02 OR 1.70 (1.03–2.8)
Mortality 90 days	52/118 44%	38/61 62.2%	125/348 35.9%	*p* = 0.17/ *p* = 0.02 OR 1.73 (1.1–2.73)
Rehospitalization	31/118 26.2%	15/61 24.5%	85/348 24.4%	*p* = 0.09/ *p* = 0.54
Intensive care unit	5/118 4.2%	1/61 1.6%	6/348 1.7%	*p* = 0.12/ *p* = 0.67
Colectomy	1/118 0.8%	1/61 1.6%	5/348 1.4%	*p* = 0.35/ *p* = 0.42
Recurrent CDI	12/118 10.1%	1/61 1.6%	21/348 6.0%	*p* = 0.08/ *p* = 0.08
Length of stay	19.1 ± 31	20.5 ± 20.3	20.1 ± 20.5	*p* = 0.65/ *p* = 0.77
Diarrhea Duration	3.83 ± 4.3	5.36 ± 5.7	5.38 ± 6.6	*p* = 0.08/ *p* = 0.89

CDI-Clostridioides difficile infection.

**Table 3 jcm-10-02957-t003:** Multivariable analysis for mortality.

	Odds Ratios
Thrombocytopenia (<150,000/mL)	1.70 (1.01–2.89)
Thrombocytosis (>450,000/mL)	1.89 (1.01–3.52)
Age	1.039 (1.02–1.057) *
ADL status–Dependent	1.71 (1.05–2.79)
Hypoalbuminemia (<3.0 mg/dL)	3.02 (1.99–4.59)
Acute kidney injury	1.34 (1.12–1.68)

* Per each year. ADL–activities of daily living.

## Data Availability

Data available on request due to restrictions eg privacy or ethical.

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
