# Peer review of "The Importance of Abnormal Platelet Count in Patients with Clostridioides difficile Infection"

_jcm, 2021, doi:10.3390/jcm10132957_

Round 1

Reviewer 1 Report

The authors of the manuscript, “The importance of abnormal platelet count in patients with Clostridioides difficle infection” have described a possible association between thrombocytosis and thrombocytopenia with CDI and increased risk of mortality. The study is well conducted, and statistical analysis are also appropriate. While the associations discussed in the manuscript are interesting, they are not entirely unexpected. It would be nice if authors add a brief section in the discussion about the relationship between the diseased conditions and treatment regimens and how these may increase the risk of thrombocytosis and thrombocytopenia as well as CDI, and eventually result in an unfavorable outcome. In addition, the Introduction can be improved by the addition of more background information on CDI etiology.

Author Response

We thank the editor and the reviewers for the chance to revise our manuscript. We had answered the reviewers’ remarks, point by point and revised accordingly. Through the revision process, we had discovered an error at the methods section under statistical analysis. This error was corrected (see below) and the manuscript was revised accordingly. Among the changes in this revised manuscript, we removed misprints, corrected copy errors and redesigned the tables. We believe that the manuscript following the revision is now clearer, and the results are better explained.

Sincerely

Dr. S. Buchrits

The authors of the manuscript, “The importance of abnormal platelet count in patients with Clostridioides difficle infection” have described a possible association between thrombocytosis and thrombocytopenia with CDI and increased risk of mortality. The study is well conducted, and statistical analysis are also appropriate. While the associations discussed in the manuscript are interesting, they are not entirely unexpected.

  • It would be nice if authors add a brief section in the discussion about the relationship between the diseased conditions and treatment regimens and how these may increase the risk of thrombocytosis and thrombocytopenia as well as CDI, and eventually result in an unfavorable outcome.

As requested, we added further data on the discussion section as follows: “Patients suffering from malignancy are at increased incidence of CDI, with increased rates of mortality, clinical failure and rCDI, length of hospitalization [17]. This is attributed to several mechanisms which include: increased exposure to broad-spectrum antibiotics; chemotherapy regimens may act as antibacterial [18], 5-fluorouracil inhibits proliferation as aerobic intestinal bacteria [19], direct mucosal toxicity and impaired repair ability with topoisomerase inhibitors [18] and platinum-based agents [19]; older age [20] and usage of proton pump inhibitors [21].”, and the associated references.

  • In addition, the Introduction can be improved by the addition of more background information on CDI etiology.

As requested, we elaborated in the introduction section on the etiology of CDI as follows: “Older age and increased comorbidity are risk factors for CDI. The rate of CDI among patients older than 65 years is 20 times higher than in younger individuals. Increased comorbidity is also related with CDI, partly by its association with greater contact with health-care and need for hospitalization. [3].” We added the associated reference.

Reviewer 2 Report

The authors of the manuscript entitled “The importance of abnormal platelet count in patients with 2 Clostridioides difficile infection” analyzed data of 527 patients all suffering from C.Diff infections with a focus on platelet counts

The paper is overall well written and of good language, style and structure. However, I have several main points regarding the findings in this study:

  • The authors associate outcome parameters with platelet counts and state, that patients with thrombocytopenia and thrombocytosis are performing worse than patients with normal platelet counts. The criteria of Thrombocytopenia/Thrombocytosis are defined over a time period of 12 months. It is unclear why a period of 1 year should have any association with the acute CDI infection. Later the authors talk about platelet counts at admission (at CDI infection?). This remains unclear to me
  • The main differences in platelet counts, especially of thrombocytopenia, seem to be driven by patients with (hemato-oncological) malignancies and this might presumably the main driver for the worse outcome of at least thrombocytopenic patients, which the authors also state in their discussion. Malignancies/chemotherapy patients should either be excluded or carefully analyzed
  • There is not data on the univariate analysis on primary endpoints that lead to the established multivariate cox model. Furthermore, what was the criteria of being included into the cox regression, p<0.1? This data should be included
  • It is unclear what the p-value in table 2 refers to: comparison of Thrombocytopenia vs. control, Thrombocytosis vs. control or Thrombocytopenia vs. Thrombocytosis?
  • It is unclear what the cause of death was, especially in the 30-day follow up period
  • It seems not appropriate to compare the survival data by chi-square tests, it is better to compare those by a survival analysis (e.g. with the Kaplan-meier Method)
  • There is no explanation on the mechanisms why patients with CDI and thrombocytosis should perform worse, association with heart problems (as associated by the author’s data)? Any recent bleeding/surgery history in patients with thrombocytosis? Splenectomy patients enriched in the thrombocytosis group?
  • There are missing table legends.

Author Response

We thank the editor and the reviewers for the chance to revise our manuscript. We had answered the reviewers’ remarks, point by point and revised accordingly. Through the revision process, we had discovered an error at the methods section under statistical analysis. This error was corrected (see below) and the manuscript was revised accordingly. Among the changes in this revised manuscript, we removed misprints, corrected copy errors and redesigned the tables. We believe that the manuscript following the revision is now clearer, and the results are better explained.

Sincerely

Dr. S. Buchrits

The authors of the manuscript entitled “The importance of abnormal platelet count in patients with 2 Clostridioides difficile infection” analyzed data of 527 patients all suffering from C.Diff infections with a focus on platelet counts. The paper is overall well written and of good language, style and structure. However, I have several main points regarding the findings in this study:

  • The authors associate outcome parameters with platelet counts and state, that patients with thrombocytopenia and thrombocytosis are performing worse than patients with normal platelet counts. The criteria of Thrombocytopenia/Thrombocytosis are defined over a time period of 12 months. It is unclear why a period of 1 year should have any association with the acute CDI infection. Later the authors talk about platelet counts at admission (at CDI infection?). This remains unclear to me.

We agree with the reviewer's remark, this sentence was accidently leftover from previous versions of the protocol / manuscript. Platelet counts were not defined over a 12 month period. We referred in our manuscript only to the actual platelet count at admission. We therefore removed that sentence. We apologize for this mistake  and the confusion.

  • The main differences in platelet counts, especially of thrombocytopenia, seem to be driven by patients with (hemato-oncological) malignancies and this might presumably the main driver for the worse outcome of at least thrombocytopenic patients, which the authors also state in their discussion. Malignancies/chemotherapy patients should either be excluded or carefully analyzed

As requested, we performed a sensitivity analysis of our primary outcome after excluding all patients with hematological malignancy, stem cell transplantation and patients under active chemotherapy.

We added the following to the methods section: ”As a sensitivity analysis, we excluded patients with hematological malignancy, stem cell transplantation and patients under active chemotherapy and repeated our analysis for the primary outcome”. We also added the following to the results section, we: “A sensitivity analysis restricted to patients without hematological malignancy, stem cell transplantation and patients under active chemotherapy, revealed mortality rates of 37.3% (83/222) for control, 40%, 19/47 for thrombocytopenia group (p=0.07) and 50% 29/58 for the thrombocytosis group (p=0.02).” We finally added the following remark in the discussion section: ”However, this might be attributed to the severity of their comorbidities rather than that of the CDI, as suggested by our sensitivity analysis (excluding patients with disease / chemotherapy related platelets count abnormalities), showed a non-significantly increased risk of mortality with thrombocytopenia (37% vs. 40%, p=0.07).” This was also added to the abstract as follows: “A sensitivity analysis restricted for patients without hematological malignancy or receiving chemotherapy revealed increased mortality with thrombocytosis but not with thrombocytopenia”.

  • There is not data on the univariate analysis on primary endpoints that lead to the established multivariate cox model. Furthermore, what was the criteria of being included into the cox regression, p<0.1? This data should be included

As requested, we rephrased the sentence on the choice of variables on the univariable analysis in the methods section as follows:” Thrombocytopenia, thrombocytosis and variables significantly associated to the primary outcome on univariable analysis, or deemed clinically relevant were entered into a multivariable logistic regression “

  • It is unclear what the p-value in table 2 refers to: comparison of Thrombocytopenia vs. control, Thrombocytosis vs. control or Thrombocytopenia vs. Thrombocytosis?

The p value always represented the comparisons between the thrombocytopenic / thrombocytosis group with the control. We rephrased the following sentence on the methods sections: “Data were compared between the control group and thrombocytopenia and thrombocytosis separately.” We separated the p value on table 1 / 2 for patients with thrombocytopenia / thrombocytosis, respectively.

  • It is unclear what the cause of death was, especially in the 30-day follow up period

Our primary endpoint was all-cause mortality at 30 days. We chose this because it is considered an indisputable, objective 'hard ' outcome in trials of infectious diseases.  All‐cause mortality encompasses all types of death: infection –related mortality and other non-infection-related causes of death) possibly related to the comorbid conditions). In addition, it accounts for the personal harm associated with antibiotic administration, side effects, and the possibly emergence of resistant micro‐organisms. We did not collect infection-related mortality or other causes of death, because infection-related mortality is a problematic outcome. The ability of physician or researchers to properly assign the cause for death in patients with infections, might be faulty. This is true in prospective trials and even more in retrospective studies, in which outcome assessors may have mistaken the cause for mortality. Therefore, all-cause mortality was chosen as the outcome.

  • It seems not appropriate to compare the survival data by chi-square tests, it is better to compare those by a survival analysis (e.g. with the Kaplan-meier Method)

We sincerely apologize for a mistake on the methods section. On previous protocols and drafts we suggested to use Cox regression multivariable analysis and to calculate Hazard ratios, however we had performed a multivariable logistic regression and odds ratios were calculated. We therefore corrected that mistake on the methods section.

Since our outcome of interest (effect of abnormal platelet count on mortality) is assumed to be constant with respect to time and we mainly aim to assess for association rather than temporal progression we used logistic regression. We therefore felt that Kaplan-Meier curve was not the statistical analysis of choice in this case. We again apologize for this error.

  • There is no explanation on the mechanisms why patients with CDI and thrombocytosis should perform worse, association with heart problems (as associated by the author’s data)? Any recent bleeding/surgery history in patients with thrombocytosis? Splenectomy patients enriched in the thrombocytosis group?

We do not have enough data to explain the specific mechanisms affecting the performance of CDI patients with thrombocytosis. We collected many parameters regarding demographics and previous conditions (Malignancy, Charlson’s comorbidity score, heart failure etc.) The only parameter that was significantly associated with thrombocytosis was ischemic heart disease. Although we show an association with ischemic heart disease, we cannot assume causality since this is retrospective. Moreover, as previously explained we did not collect data for causes of death. We do not have data regarding splenectomized patients, or previous bleeding history. As we elaborate in the discussion, we assume the mechanisms affecting the performance of CDI patients with thrombocytosis may probably be attributed to the severity of the disease and the inflammation (since the platelet is regarded as an acute phase reactant), as in many other infectious/inflammatory situations. We added a sentence explaining this to the discussion section